# A Study of Patients’ Willingness to Pay for a Basic Outpatient Copayment and Medical Service Quality in Taiwan

**DOI:** 10.3390/ijerph18126604

**Published:** 2021-06-19

**Authors:** Wei Hsu, Chih-Hao Yang, Wen-Ping Fan

**Affiliations:** 1Department of Health Care Management, National Taipei University of Nursing & Health Sciences, Beitou, Taipei 112303, Taiwan; tunghui@thltc.com; 2Department of Accounting, Ming Chuan University, Shilin, Taipei 111005, Taiwan; chihhao@mail.mcu.edu.tw

**Keywords:** willingness to pay, medical service quality, basic outpatient copayment, hierarchy of medical care

## Abstract

Patients might be willing to pay more to obtain better quality medical services when they recognize that high-level hospitals have better quality. However, published papers have not found solid empirical evidence to support this possibility. Therefore, the purpose of this study is to empirically investigate patients’ willingness to pay (WTP) for an outpatient copayment. The study aims to analyze the difference between the two WTP values: to implement a hierarchy of medical care and to improve the quality of medical services. This study administered a questionnaire using the contingent valuation method with a quasi-bidding game for patients’ WTP and the SERVQUAL scale for medical service quality. The Wilcoxon signed-rank test was employed to test the difference between the two WTP values, notably to implement a hierarchy of medical care and to improve the quality of medical services. Both of the WTP values are higher than the academic medical centre’s current copayment NT$420 (approximately US$14); the percentage of respondents willing to pay a higher copayment declined when the outpatient copayment was increased, and the patients’ WTP to have better medical service quality was significantly higher than that to implement a hierarchy of medical care. Patients’ desire to receive better medical services from higher-level hospitals might be stronger than their desire to implement hierarchical medical care. This study reported the relationship between the respondents’ perceived medical service quality and WTP for having better service quality by using regression models. The respondents’ perceptions of medical service quality, especially for “reliability” and “assurance,” would positively affect their WTP. Policy makers should focus on improving the quality of medical services.

## 1. Introduction

Since the National Health Insurance (NHI) system of Taiwan was created in 1995, the NHI of Taiwan provides medical services with acceptable quality levels; however, the utilization rate of medical services has grown rapidly. The Ministry of Health and Welfare of Taiwan adopted a copayment system for outpatient care to eliminate the waste of medical resources. To implement a hierarchy of medical care, copayments vary across different levels of medical institutions. The hierarchy of medical care in Taiwan includes four levels. For medical centers, the outpatient copayment without referral rose from NT$210 (approximately US$7) in 1995 to NT$360 (approximately US$12) in 2005 and NT$420 (approximately US$14) in 2017; for regional hospitals, outpatient copayment without referral was from NT$150 (approximately US$5) in 1995 to NT$240 (approximately US$8) in 2005. The outpatient copayment of district hospitals without referral was NT$80 (approximately US$3) and the outpatient copayment of general practice clinics was NT$50 (approximately US$2) from 1995 until now. However, for higher-level hospitals, such as medical centers and regional hospitals, the number of medical expenditure applications for outpatient services has increased rapidly in recent years, but the numbers for district hospitals and general practice clinics have decreased.

Citizens have gained access to an acceptable quality of medical services; however, medical expenditure applications for outpatient services at hospitals have increased rapidly in recent years. In particular, the number of medical expenditure applications for outpatient services from district hospitals and general practice clinics decreased, while the number of applications from academic medical centers and regional hospitals increased. Healthcare authorities in Taiwan created a hierarchy of medical care and repeatedly adjusted outpatient copayments to reduce the waste of medical resources, especially through outpatient copayments in higher-level medical institutions. However, the number of medical expenditure applications for outpatient services from high-level medical institutions has not decreased but increased. Since that demand for medical services is inelastic and information asymmetry exists between doctors and patients, patients might not consciously reduce outpatient visits to higher-level hospitals in response to a higher copayment for outpatient visits at these hospitals or government advocacy of hierarchical medical care. Increased copayments might be able to reduce the number of outpatient visits only for a short time and then cease to be effective at reducing the medical utilization rate.

Some previous studies have investigated willingness to pay (WTP) and the quality of medical services and stated that higher-level hospitals might have better quality and that patients thus have a higher WTP copayments. People who can afford to do so should be willing to pay more to obtain better quality medical services [1]. Patients are willing to pay for quality improvements in public health care services [2]. Analysis of different levels of doctors reveals that people have higher WTP for higher-level doctors [3]. In addition, patients are willing to pay more if they obtain a closer doctor-patient relationship, increased drug availability and improved the chances of recovery. Moreover, the doctor-patient relationship is regarded as the most important factor and has shown a high WTP [4]. The patients regard a policy change as suitable if the payment is reinvested in equipment to improve quality and service access [5]. Moreover, those who can afford a higher payment might not be willing to accept the current expected service quality, and they might spend more to obtain better service quality [5,6].

The quality of a large institution with professional equipment will be higher than that of a general hospital [7]. Large hospitals with a large number of beds and teaching hospitals have the highest disease improvement rate and best predictive indicators. Consequently, many quality predictors for treatment of important diseases are based on large hospitals. Academic medical centers are more efficient than regional and district hospitals, which can be attributed to their scale and scope [7]. The limited resources of small hospitals entail lower medical service quality [8]. Patients will spend more time and money obtaining high-quality services from their ideal medical institutions instead of lower-level hospitals [9,10,11]. However, research has revealed that older individuals with noninfectious or chronic diseases can receive sufficient care for their common health needs in primary medical institutions. Therefore, there is considerable potential for increasing the efficiency with which health resources are utilized, and the referral system and hierarchy of medical care would be improved by doing so [12].

The purpose of implementing a copayment system is to make patients aware of medical resources and in turn promote health and cost awareness. The aim is to urge patients to be consumers who understand the concept of medical costs under the copayment system [2]. Research reveals that most hospital claims in high-level hospitals are attributable to simple diseases and chronic patients [13]. In addition, some studies have reported that adjusting the prescription drug copayment can effectively induce chronic patients to use clinics to manage their diseases. Moreover, under current policy, patients with chronic and mild diseases impose a more significant cost burden than acute patients [14,15]. However, a higher copayment causes low-income people and patients with chronic disease to reduce their use of primary care, which might result in worse health. This would in turn lead to high-cost treatment [16]. As reviewed previously, it could be necessary to ensure high-quality services and charge an appropriate copayment to encourage chronic and mild patients to use the appropriate hospital levels [14].

The WTP represents a specific price that consumers are willing to pay for a product [17]. The principle of the contingent valuation method (CVM) is to use different methods to directly ask respondents and then derive their WTP for non-market goods. The research tools of the CVM include open-ended, close-ended, payment card, bidding game [18,19] and quasi-bidding game [20] approaches. Several published studies have employed the CVM to access people’s WTP for health insurance services [21,22,23,24]. The topics include health insurance [20], diagnostic technology [25], quality adjusted life years [26], and other health care services [27,28,29,30,31]. However, none of these studies employed the same method to investigate WTP. The method of investigation should be selected based on the characteristics of the problem at hand. The double-bounded dichotomous choice and the bidding game approaches usually suffer from starting point bias; the initial reference price usually affects the overall WTP. Using payment cards introduces the problem of price range error. The range of reference prices also affects the estimation of the overall WTP [32]. The quality of the CVM depends on the simulation and reliability of the hypothetical. There are 5 key points in developing hypotheticals, including context establishment, how to charge, whether the goods are risky or uncertain, the period of valuation and how to investigate. Therefore, it is necessary to clearly understand the characteristics of the goods and the content to avoid obtaining unclear results [33]. Studies have investigated the WTP for medical insurance in developing countries with a low insurance coverage rate [24] or sampled respondents without medical insurance in developed countries [22]. In contrast to the case of the countries examined in previous studies, the coverage rate of NHI, as compulsory social insurance in Taiwan, is higher than 99%. Thus, this study employed the CVM with a quasi-bidding game to empirically assess the WTP for an outpatient copayment in Taiwan to analyze the differences between implementing hierarchical medical care and improving the quality of medical services. In summary, the CVM with a quasi-bidding game has been widely used to investigate WTP in the health care industry.

Published papers have not found solid empirical evidence to support the idea that higher-level hospitals which have better quality would lead to a higher patient WTP for copayments. Therefore, the purpose of this study is to explore the WTP of outpatient copayments by using CVM with a quasi-bidding game. It aims to empirically investigate patients’ WTP for an outpatient copayment and analyze the difference between the WTP to implement a hierarchy of medical care and the WTP to increase the quality of medical services. This study would also report the influence of the respondents’ perceived medical service quality to WTP. The respondents almost certainly understand the subject because outpatient copayments have been in place for many years in Taiwan. To prevent respondents from selecting any given price, the former adjustment range was used as a starting price for reference. This study uses a quasi-bidding game, which limits the range of price options, and employs open-ended games to obtain the maximum WTP in outpatient copayment.

## 2. Research Methods

This study is based on the relevant literature to establish a variable measurement method. A semi-structured questionnaire is used as the main research tool. The content of the questionnaire is divided into three parts: medical service quality, WTP and basic information on the respondents.

The questionnaire in this study is modified from the SERVQUAL questionnaire [34] to fit the purpose of this study. The section of the questionnaire on medical service quality included five dimensions (tangibility, reliability, responsiveness, assurance and empathy) and 28 questions measured on a seven-point Likert scale (1 = strongly disagree, 2 = disagree, 3 = somewhat disagree, 4 = neutral, 5 = somewhat agree, 6 = agree, 7 = strongly agree).

WTP is a specific price that consumers are willing to pay for a product [17]. Health insurance represents a non-market good [21], and copayments are also paying for non-market services, so this study employed the CVM to investigate the value of non-market goods. To make sure respondents realized the amounts of the current outpatient copayment without referral, the WTP questionnaire also presented the current outpatient copayment without referral for the four levels of hierarchy of medical care in Taiwan (NT$360 (approximately US$12) of medical centers, NT$240 (approximately US$8) of regional hospitals, NT$80 (approximately US$3) of district hospitals, NT$50 (approximately US$2) of general practice clinics). According to the quasi-bidding game of CVM, as shown in Figure 1, this study designed two separate games that started from two different questions, “Would you willing to pay more than NT$360 of outpatient copayment to implement hierarchy of medical care?” and “Would you willing to pay more than NT$360 of outpatient copayment to have better medical service quality?” If the answer was “Yes,” the respondent would continue to the Part II; if the answer was “No,” the maximum amount of WTP would be recorded as NT$360. Then, in the second part of the WTP section (Part II) of the questionnaire, this study offered closed-ended single-bounded dichotomous choices to investigate respondents’ WTP. In Part II of both games, the amount of outpatient copayment would be set as NT$440 (higher than NT$360) and the respondent would be asked if he/she would be willing to pay. If the answer was “Yes,” the respondent would be asked if he/she would be willing to pay for NT$460 (higher than NT$440); if the answer was “No,” the respondent would be asked if he/she would be willing to pay for NT$420 (lower than NT$440). All respondents who answered Part II of the WTP questionnaire would be requested to continue to increase (or decrease) the amount of WTP and finally to report their maximum amount in an open-ended question in Part III of the WTP questionnaire. In the third part, the closed-ended double-bounded dichotomous choice approach was used to let the respondents report the value of their WTP.

At the end of the questionnaire, six items were included to elicit basic demographic information on the respondents: gender, age, education, occupation, monthly disposable income and marital status.

In designing the questionnaire for this study, each scale was based on theoretical constructs and had nomological validity. In addition, the content validity of the questionnaire was ensured by consulting four experts (a senior physician and three professors in related fields) who have rich experience in medical health care, service quality and economics. Cronbach’s α was calculated to test the internal consistency of the five dimensions of medical service quality. The Cronbach’s α of all dimensions is higher than 0.8, which indicates that the questionnaire had good internal consistency. The questionnaire for this study was approved by the Institutional Review Board (IRB No.: 2017-03-001AE).

The survey was administered from 29 December 2016 to 21 January 2017. To avoid resampling, the sampling periods included morning, afternoon and night from Monday to Friday, and Saturday morning (a total of 16 periods). To ensure sample homogeneity with social background in 2015, all subjects’ gender and age distributions were tested using the Chi-square test. The test results reported in Table 1 indicate that there was no significant difference in gender between the sample and the population (χ^2^ = 0.84, *p* = 0.361) and no significant difference in age between the sample and the population (χ^2^ = 1.86, *p* = 0.602). This result indicates that the sample considered in this study accords with the characteristics of the 2015 NHI-insured population in terms of gender and age.

The purpose of this study is to investigate whether there is a difference between the WTP for an outpatient copayment to implement hierarchical medical services and improve the quality of medical services, and the influence of the respondents’ perceived medical service quality on WTP. The variables to be tested in this study are continuous variables. WTP values in the literature are mostly right-skewed. If the distribution of the WTP in this study is also right-skewed, then the nonparametric statistics of the Wilcoxon signed-rank test should be used. The Wilcoxon signed-rank test analyses the difference in the medians of sets of observations. It is suitable for continuous variables and does not require the researcher to assume that the data are normally distributed. Therefore, in order to analyze the relationship between the respondents’ perceived medical service quality and WTP, the multiple linear ordinary least squares (OLS) regression model was used, and this study employed bootstrap as this technique does not rely on any distribution assumption but uses the observed distribution for estimating standard errors [35].

## 3. Research Results

In order to avoid that the respondents might feel confused by the different amounts of outpatient copayments in the different levels of hospitals, the research object of this study focused on patients of academic medical centers and did not investigate other levels of the hospital hierarchy since the NHI of Taiwan has already adjusted the amount of outpatient copayments without referral for medical centers three times from 1995 until now. This study sampled patients from an academic medical Center in Taipei, Taiwan, and collected a total of 1134 questionnaires during the survey period. Because all of the respondents finished the questionnaire in this medical Center, so the precondition is that they are already in the academic hospital. After screening and filtering invalid questionnaires (respondents who were under 18 years old or qualified for a reduced outpatient copayment, such as disabled persons, patients with major illnesses and veterans), there were 971 valid questionnaires, and the questionnaire recovery rate was 85.63%.

### 3.1. Demographic Analysis

Table 2 presents the demographic analysis of this study. There are six demographic variables: gender, age, education, occupation, monthly disposable income and marital status. In terms of respondents’ gender, the proportion of men and women was quite similar, approximately 51% men and 49% women. In terms of the respondents’ age distribution, approximately 23% were younger persons aged between 18 and 30 years, 26% were members of the middle-aged population aged between 31 and 44 years, 36% were members of the middle-aged population aged between 45 and 64 years, and 14% were elderly persons aged over 65 years. According to the results of the Chi-square test, there was no significant difference in gender and age between the sample and the real population of Taiwan in 2015, which shows that the sample was representative.

The majority of the respondents (56.85%) had a college-level education, followed by high school (25.75%) and graduate school (17.20%). In terms of the respondents’ occupation, general private employees accounted for the highest proportion (39.55%), followed by the self-employed (18.33%) and military/civil servants/teachers (8.03%). In terms of monthly disposable income, 19% of respondents had more than NT$50,000 (approximately US$1663) per month. In terms of marital status, the majority of respondents were married (58.81%). As monthly disposable income per household and household population are open-ended questions in the questionnaire, monthly disposable income cannot be calculated when either of these two responses is missing, and because income issues are relatively sensitive, respondents often refuse to answer. This resulted in more missing data on monthly disposable income (12.67%).

### 3.2. Medical Service Quality and Willingness to Pay

The respondents’ perceived medical service quality and WTP are shown in Table 3 and Table 4. In terms of the overall medical service quality score, respondents perceived a positive medical service quality for the target academic medical Center in Taipei, with an average score of 5.77. The mean values of perceived medical service quality on tangibility, reliability, responsiveness, assurance and empathy were all greater than 3.5, with the mean values ranging from 5.64 to 5.90. The percentage of missing data on the five dimensions of medical service quality ranged between 2% and 4%, while the share of missing data on the overall medical service quality was less than 10%. According to Table 4, there were 665 respondents (68.49%) who reported being willing to increase the outpatient copayment to implement hierarchical medical care, and the average WTP was NT$469.63 (approximately US$16); 740 respondents (76.21%) would be willing to pay more to improve the quality of medical services, and the average WTP was NT$510.42 (approximately US$17).

Furthermore, to obtain a detailed understanding of the changes in the proportion of respondents who would be willing to pay a higher outpatient copayment, this study calculated the numbers and proportions of respondents who would be willing to increase WTP in Table 5 and drew two charts in Figure 2 for two different purposes. The x-axis of the two charts represents the different levels of WTP; the y-axis represents the proportion of respondents who would be willing to increase WTP. From chart (a) of Figure 2, regarding the implementation of hierarchical medical care, 61.42% of respondents were willing to pay a higher outpatient copayment of NT$420 (approximately US$14), 54.43% were willing to increase the copayment to NT$440 (approximately US$15), and 46.63% were willing to increase the copayment to NT$460 (approximately US$16). From chart (b) of Figure 2, regarding improving the quality of medical services, 69.05% of respondents were willing to increase and pay a higher outpatient copayment of NT$420 (approximately US$14), 62.33% of respondents were willing to pay NT$440 (approximately US$15), and 55.45% of respondents were willing to pay NT$460 (approximately US$16). The proportions of respondents who were willing to pay more to implement hierarchical medical care and for improve the quality of medical services all decreased with increasing WTP levels. However, the proportions of respondents who were willing to pay to have better medical service quality were higher than the proportions of respondents who were willing to pay to implement hierarchical medical care in each level of WTP.

In this study, the total average (5.77) of all respondents’ perceived medical service quality was taken as the threshold, and the respondents were divided into two groups: those with lower perceived medical service quality (≤5.77) and those with higher perceived medical service quality (>5.77). The numbers and proportions of respondents at the different levels of WTP for the two groups of perceived medical service quality are shown in Table 6. Overall, 68.52% of respondents who had lower perceived medical service quality were willing to increase the payment to NT$420 (approximately US$14) to have better medical service quality, 60.05% were willing to increase the payment to NT$440 (approximately US$15), and 53.21% were willing to increase the payment to NT$460 (approximately US$16). Of the respondents who had higher perceived medical service quality, 70.35% were willing to increase the payment to NT$420 (approximately US$14), 65.03% were willing to increase the payment to NT$440 (approximately US$15), and 57.61% were willing to increase the payment to NT$460 (approximately US$16).

Regardless of whether one considers the respondents with lower or higher perceptions of medical service quality, the numbers and proportions of respondents who were willing to pay decreased when the level of WTP to improve the quality of medical services increased. The proportions of respondents who had high perceptions of medical service quality and were willing to pay were higher than the proportions of respondents who had low perceptions of medical service quality and were willing to pay at each level of WTP to have better medical service quality.

### 3.3. Wilcoxon Signed-Rank Test and Regression Analysis

This study also tested whether there is a significant difference between the WTP for an outpatient copayment to implement hierarchical medical care, and to improve the quality of medical services. Before conducting the test, it is necessary to identify the distribution of the WTP for an outpatient copayment, as shown in Figure 3. There is a right-skewed distribution of the WTP for an outpatient copayment both to implement hierarchical medical care and for having better medical service quality. Therefore, the Wilcoxon sign-rank test was chosen to further test whether a significant difference exists between the medians of the WTP for an outpatient copayment to implement hierarchical medical care and to have better medical service quality.

The results of the Wilcoxon sign-rank test are shown in Table 7. There was a statistically significant difference between the WTP values for implementing hierarchical medical care and for having better medical service quality (*z* = 10.19, *p* < 0.001). The WTP to have better medical service quality is significantly higher than that to implement hierarchical medical care.

Furthermore, this study reported the relationship between the respondents’ perceived medical service quality and WTP for having better service quality by ordinary least squares (OLS) regression models. As shown in Figure 3, there is a right-skewed distribution of the WTP for having better medical service quality, so this study employed bootstrap (2000 replications). The results showed that the respondents’ perceived medical service quality would significantly and positively affect the WTP for having better service quality (β = 5.42, *p* < 0.001). When the respondents’ perceived medical service quality is improved one level higher (such as from “neutral” to “somewhat agree”), the WTP for having better medical service quality would increase by NT$5.42 (approximately US$0.18). This study also found that the two dimensions of service quality (“reliability” (β = 8.68, *p* < 0.001) and “assurance” (β = 15.68, *p* < 0.001)) have especially strong influences on the WTP for having better medical service quality.

## 4. Conclusions

### 4.1. Discussion and Implication

The purpose of this study is to investigate the willingness of Taiwanese people to pay an outpatient copayment to implement hierarchical medical care and to have better medical service quality, as well as the difference between the WTP to implement hierarchical medical care and to have better medical service quality. The results of this survey show that the average medical service quality score of the target academic medical center was 5.77, indicating that the public strongly approved of the quality of service provided by the academic medical center, especially on the two dimensions of “reliability” and “assurance” (which were between 5.87 and 5.90 on average). The results showed that the majority of the respondents believed that the academic medical center provided good service; that is, the service actually provided by the staff of the academic medical center was as promised and that the staff members were polite, knowledgeable and trustworthy.

This study also found that the mean of respondents’ WTP as an outpatient copayment to implement hierarchical medical care was NT$470 (approximately US$16), and that to have better medical service quality was NT$510 (approximately US$17). Both means of respondents’ WTP were higher than the actual outpatient copayment at academic medical centers at the time of the survey (the end of 2016), which was NT$360 (approximately US$12), and higher than the outpatient copayment of NT$420 (approximately US$14), as adjusted on April 15, 2017. From 1995 until now, the NHI of Taiwan increased the outpatient copayment without referral for medical centers three times (from NT$210, NT$360 to NT$420) to enlarge the differences between medical centers and the other three levels of hospitals, but this did not have obvious effects on patients’ hierarchical medical behaviors.

According to the results of the Wilcoxon sign-rank test, the WTP to have better medical service quality was significantly higher than that to implement hierarchical medical care. A possible implication of this result is that patients’ desire to receive better medical services from higher-level hospitals might be stronger than their desire to implement hierarchical medical care. People’s incentives to have better medical service quality would be stronger than those to implement hierarchical medical care. From the results of regression analyses, the respondents’ perceived medical service quality, especially on “reliability” and “assurance,” positively affects their WTP. Therefore, policy makers seeking to adopt outpatient copayments to implement hierarchical medical care face a difficult task. Policy makers might focus on improving the quality of medical services in lower-level hospitals. When people believe that lower-level hospitals can provide the same quality as higher-level hospitals, they will be willing to implement hierarchical medical care. Not only promoting the concepts of hierarchical medical care and increasing the outpatient copayment without referral for medical centers, policy makers for the long-term should also consider how to invest more resources to improve medical service quality in lower-level hospitals, especially in “reliability” and “assurance” of medical service quality.

### 4.2. Limitations

This study, due to the limited time, funds and other conditions, only distributed questionnaires at a target academic medical center in Taipei, Taiwan. Although there might be regional differences, the distributions of the gender and age of the academic medical center patients were similar to the distributions of gender and age for all Taiwanese citizens in the NHI, and the results of this study thus have some degree of external validity. However, outpatient copayments in Taiwan differ across different levels of hospitals, and this study did not investigate other levels of the hospital hierarchy to avoid the respondents feeling confused by the different amounts of outpatient copayments in the different levels of hospitals. This could cause a possible selection bias, so the WTPs might be overestimated. Future studies could extend the present work to a comparative analysis of different levels of hospitals.

In this study, the CVM was adopted to obtain the WTP for outpatient copayments. Compared with the payment card method, the CVM can improve the accuracy of WTP values [21] but might also be affected by the setting of the inquiry point. Therefore, the WTP levels before and after the adjustment of the copayment might be different. Future studies could explore whether there is a difference in the WTP in the short term before and after partial copayment adjustment to eliminate the bias caused by the impact of government policy. In addition, respondents might be concerned that reporting a higher WTP will affect the government’s adjustment policy for outpatient copayments; thus, the WTP values in this study might be underestimated. In other words, the WTP in this study might be affected by response bias arising from the respondents’ strategic behavior [21,36].

This study focused on the WTP the non-referral outpatient copayment at an academic medical center and did not consider the effect that might arise from the incentives from referrals, which have a lower copayment. Future studies could be expanded to consider outpatient copayment for referrals and analyze patients’ willingness to accept an outpatient copayment for referrals. Moreover, on 15 April 2017, the Taiwanese government raised the outpatient copayment for Western medicine outpatient services without a referral to NT$420 (approximately US$14), which is still much lower than the WTP value for the outpatient copayment investigated in this study. Future studies could further explore whether the adjustment of the outpatient copayment affects people’s actual medical behavior and the relationship between patient’s WTP and their total cost per visits.

## Figures and Tables

**Figure 1 ijerph-18-06604-f001:**
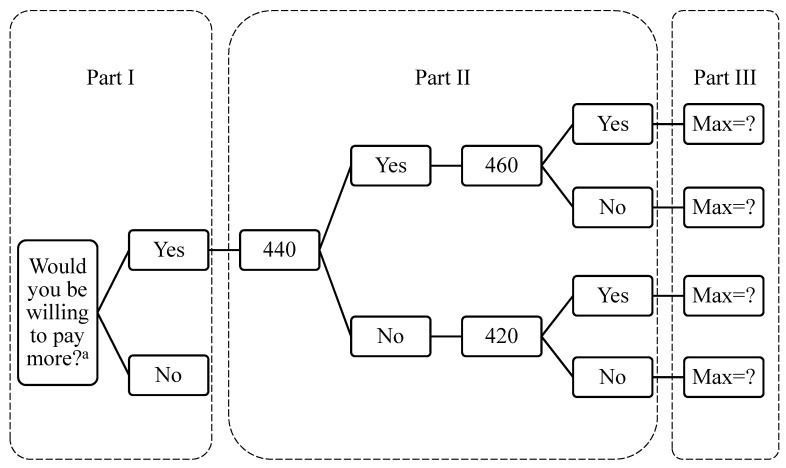
Quasi-bidding game of contingent valuation method. ^a^ There are two quasi-bidding games, “Would you willing to pay more than NT$360 of outpatient copayment to implement hierarchy of medical care?” and “Would you willing to pay more than NT$360 of outpatient copayment to have better medical service quality?”.

**Figure 2 ijerph-18-06604-f002:**
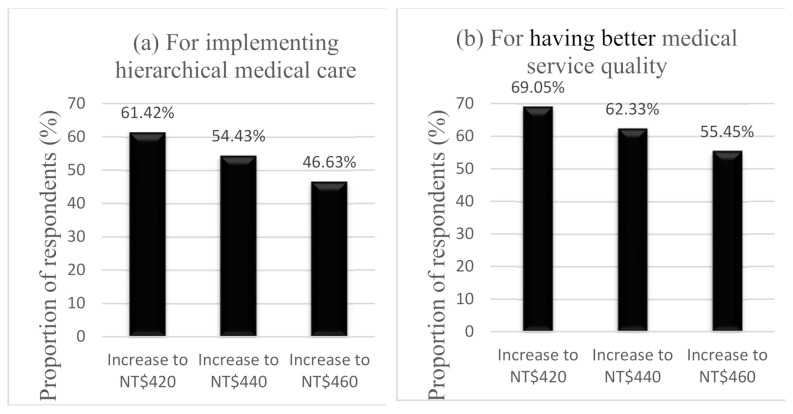
Proportion of respondents who would be willing to increase WTP.

**Figure 3 ijerph-18-06604-f003:**
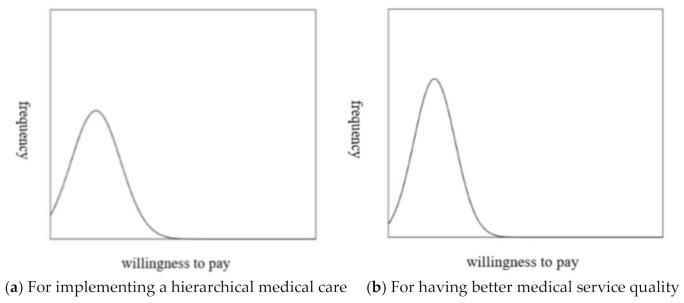
The distribution pattern of the WTP for outpatient copayment.

**Table 1 ijerph-18-06604-t001:** Chi-square fitness test results for sample proportion.

Variables	Population (%)	Sample Size (%)	Chi-Square Value	*p*-Value
Gender			0.84	0.361
Male	9,646,254 (49.10)	491 (50.57)		
Female	10,000,020 (50.90)	480 (49.43)		
Age			1.86	0.602
18~30	4,392,985 (22.36)	225 (23.17)		
31~44	5,534,748 (28.17)	257 (26.47)		
45~64	6,888,455 (35.06)	353 (36.35)		
Above 65	2,830,086 (14.41)	136 (14.01)		

Note: The population was those insured with NHI in 2015.

**Table 2 ijerph-18-06604-t002:** Characteristic Analysis (N = 971).

Variable	Frequency (%)	Missing Data (%)
**Gender**		0
Male	491 (50.57)	
Female	480 (49.43)	
**Age**		0
18–30	225 (23.17)	
31–44	257 (26.47)	
45–64	353 (36.35)	
>65	136 (14.01)	
**Education**		2 (0.21)
High school	250 (25.75)	
College	552 (56.85)	
Graduate school ^a^	167 (17.20)	
**Occupation**		7 (0.72)
Military/Civil servants/Teachers	78 (8.03)	
Private Employee	384 (39.55)	
Self-employed	178 (18.33)	
Others ^b^	324 (33.37)	
**Disposable Income per month**		123 (12.67)
<50,000	659(67.87)	
>50,000	189(19.46)	
**Marital status**		4 (0.41)
Married	571(58.81)	
Unmarried	325(33.47)	
Others ^c^	71(7.31)	

Note: ^a^ includes master’s degree and doctoral degree; ^b^ includes unemployed, housekeepers, the retired and students; ^c^ includes divorced, cohabiting and widowed.

**Table 3 ijerph-18-06604-t003:** Respondents’ medical service quality.

Items of Medical Service Quality	*n*	Missing Data (%)	Mean (SD)
Overall medical service quality	879	93 (9.58)	5.77 (0.74)
Tangibles	937	34 (3.50)	5.66 (0.86)
Reliability	933	38 (3.91)	5.87 (0.71)
Responsiveness	939	32 (3.30)	5.71 (0.87)
Assurance	940	31 (3.19)	5.90 (0.81)
Empathy	951	20 (2.06)	5.64 (1.02)

Note: *n* represents sample size, SD represents standard deviation.

**Table 4 ijerph-18-06604-t004:** The willingness to increasing outpatient copayment and average WTP.

Willingness	*n*	Missing Data (%)	*n* of Yes (%)	Average of Max WTP (SD)
For implementing a hierarchical medical	963	8 (0.82)	665 (68.49)	469.63 (207.55)
For having better medical service quality	963	8 (0.82)	740 (76.21)	510.42 (325.24)

Note: *n* represents the sample size without missing data, SD represents the standard deviation.

**Table 5 ijerph-18-06604-t005:** Numbers and proportions of respondents in the different levels of WTP.

Levels of WTP	For Implementing Hierarchical Medical		For Having Better Medical Service Quality	
	*n*	*n* of Yes (%)	*n*	*n* of Yes (%)
Increase to NT$420	959	589 (61.42)	953	658 (69.05)
Increase to NT$440	959	522 (54.43)	953	594 (62.33)
Increase to NT$460	950	443 (46.63)	945	524 (55.45)

Note: *n* represents the sample size without missing data.

**Table 6 ijerph-18-06604-t006:** Numbers and proportions of respondents in the different levels of WTP for the two groups of medical service quality.

Levels of WTP	Low Quality of Medical Service ^a^		High Quality of Medical Service ^b^	
	*n*	*n* of Yes (%)	*n*	*n* of Yes (%)
Increase to NT$420	378	259 (68.52)	489	344 (70.35)
Increase to NT$440	378	227 (60.05)	489	318 (65.03)
Increase to NT$460	374	199 (53.21)	486	280 (57.61)

Note: *n* represents the sample size without missing data, SD represents the standard deviation. ^a^ Under the average of medical service quality (≤5.77). ^b^ Over the average of medical service quality (>5.77).

**Table 7 ijerph-18-06604-t007:** Wilcoxon sign-rank test of the WTP for the outpatient copayment (*n* = 913).

	Median	Statistics	*p*-Value
For implementing a hierarchical medical care	440.00	10.19	*p* < 0.001
For improving medical service quality	460.00		

## Data Availability

The data presented in this study are available on request from the corresponding author. The data are not publicly available due to the containing information that could compromise the privacy of research participants.

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
