# Peer review of "A Study of Patients’ Willingness to Pay for a Basic Outpatient Copayment and Medical Service Quality in Taiwan"

_ijerph, 2021, doi:10.3390/ijerph18126604_

Round 1
Reviewer 1 Report
The article lacks a clearly defined aim of the research and the hypotheses / research questions posed.
In the introduction, the authors should add the characteristics of the functioning of healthcare system. For a reader not related to the country in question, it is difficult to find out in its realities, and thus in the issue of introducing copayments.
The discussion looks like a report not like a real discussion. It is definitely not enough. The authors should refer to other research results.
What are policy implications? What is the contribution of the article?
Reviewer 2 Report
This study tried to investigate patients’ WTP for an outpatient co-payment and analyse the difference between the WTP to implement a hierarchy of medical care and increase the quality of medical services. A major question is that the authors need to give more information on the study design and make the process more clear:
- One of the aim is to investigate the WTP for a hierarchy of medical care, but this study only sampled people from an academic hospital (seems top high level of hospital). There is a possible selection bias, and the WTP was overestimated.
- “Would you willing to pay more to improve the quality of medical services?”. Following comment 1, it means get more quality care in the academic hospital (the precondition is that they are already in the academic hospital) or get the more quality care by going to academic hospital (the precondition is that they are yet not in the academic hospital). What was done to ensure the samples correctly understand what you really means?
- Similar problem on “hierarchy”. It means from district hospital to regional hospital?
- The survey process in figure 1 need to be presented clearer. What the means of “440”, “460”, and “420”, and what the questions were used, respectively.
- A pity of this study is that it report the quality and WTP separately, and the results of quality do not add any new evidence. The authors should explore the relationship between quality and WTP, for instance, for one unit increase in quality how much the patients want to pay.
- It was also recommended to present the ratio of WTP and the total cost of per visiting. Only absolute value mask the information improving the policy needs.
Other comments are:
- It’s better to combine the part of introduction and review, as lots of information cannot be understood in the part of introduction but in review.
- Add discussion on how the results contribute to the practice.
Round 2
Reviewer 2 Report
The author improved the mansuscript based on my comments. No more comments.
Author Response
Dear Reviewer,
Thank you again for your recommendations. All the authors appreciate your precious comments and suggestions, and really hope our study can make a contribution to the literature.